# Insomnia in Adolescence

**DOI:** 10.3390/medsci6030072

**Published:** 2018-09-01

**Authors:** Innessa Donskoy, Darius Loghmanee

**Affiliations:** Advocate Children’s Hospital, Park Ridge, IL 60068, USA; Darius.Loghmanee@advocatehealth.com

**Keywords:** insomnia, adolescents, adolescence, teenagers, delayed sleep phase, cognitive behavioral therapy for insomnia (CBT-I)

## Abstract

Adolescent insomnia is a common condition that negatively impacts a developing young adult’s mental and physical health. While the treatment of adult insomnia has been standardized, the treatment of pediatric insomnia is very practitioner-dependent and few large-scale studies are available to determine a standard recommended practice. There is great hope that as the adolescent medicine and sleep medicine fields flourish, larger cohort analyses will be performed to determine the prevalence and precipitating factors of adolescent insomnia, allowing for standardized treatment recommendations and systematic efforts to make these recommendations available to all adolescents.

## 1. Introduction

Insomnia is a nocturnal disorder that profoundly impacts an individual’s performance during waking hours, taking away the restorative physical and cognitive properties that sleep provides [1,2]. Rather, insomnia fills this void with impatience, anxiety, and ultimately leads to maladaptive behaviors; these often involve utilization of screens, providing stimulation and light exposure, which is correlated with even later sleep onset [3]. Adolescence encompasses a wide time range which includes early (11–13 years), middle (14–18 years) and late periods (19–21 years old or through college age) [4]. Within this timeframe most individuals will go through the pubertal process, move from more logical thought to abstract thinking with problem-solving abilities, and experience a peak in impulsivity that gradually dissipates into behaviors such as self-regulation and planning [3]. This is also a time during which the adolescent will distance him or herself from the family unit and develop stronger relationships with peer groups, romantic interests, and ultimately physically separate to live independently and pursue goals such as further education or employment. This separation includes decreased sharing of medical information, even in cases of pain [5], due to embarrassment and a desire for independence. These factors do not bode well for the trend of increased issues with sleep in adolescence [6]. The amount of recommended sleep during the adolescent period is greater than expected, with the National Sleep Foundation recommending 9–11 h of sleep for children 6–13 years old and 8–10 h per night for individuals aged 14–17 years old [7]. In today’s world of increased demands on adolescents for afterschool activities to compete for limited spots in universities [8], increased exposure to screens even in their own bedrooms [7], early start times for schools [9], and a physiologic sleep/wake delay [10], there are a multitude of reasons that adolescents do not reach the critical amount of sleep they need. In fact, the National Sleep Foundation poll of 12th graders (17–18 years old) reported that 75% slept less than 8 h per night [11]; this is even more striking when considered in light of the fact that self-report of sleep by adolescents is often overestimated [12]. These adolescents are not getting adequate sleep, affecting performance during formative years [13,14], and not seeking help for it. In the setting of any underlying physiological or psychological conditions, this is often exacerbated [15,16,17]. In adults, the standard of care for insomnia has moved away from medication in recent years to more cognitive behavioral therapy for insomnia (CBT-I) [18] and while benefit has been seen in adolescents undergoing CBT-I [19,20,21] there is still limited access for those who are seeking providers of this specific technique [22]. While insomnia in the adolescent is challenging due to its multifactorial nature, it is a topic that deserves specific attention. 

## 2. Epidemiology

As early as infancy, insomnia can be found disrupting schedules, family life, and the ability for a child to have adequate daytime alertness and energy to reach their developmental goals [22]. Behavioral insomnia of infancy is a condition largely influenced by environmental factors and caregiver choices. It also provides a breeding ground for maladaptive behaviors; sleep onset associations, inappropriate limit setting, and mistiming of sleep onset time can all linger and become part of an older child’s sleep related issues. It is estimated that insomnia can be as common as 36% in preschool (3–5 year old) children and approximately 20% in school age (5–10 year old) children [23]. The prevalence of insomnia in adolescents remains in the same approximate range, as high as 23.8% [24]. This suggests that it is the most common sleep disorder in this age group. These values arise from the use of the fourth edition of the Diagnostic and Statistical Manual of Mental Disorders (DSM-IV) criteria for diagnosis of primary insomnia which is defined as at least one month of symptoms including difficulties with sleep onset, maintenance, or a sensation of unrefreshing sleep [25]. This is in contrast to fifth edition of the Diagnostic and Statistical Manual of Mental Disorders (DSM-V) which has stricter criteria for insomnia diagnosis, requiring a chronicity of 3 months [26]. In the sleep field, insomnia is diagnosed according to the third edition of the International Classification of Sleep Disorders (ICSD-3); it is described as a sleep onset or maintenance problem in the setting of an adequate opportunity to sleep, with some reported daytime impairment [27]. The causes of insomnia are multifactorial, with genetic, biologic, environmental and social factors playing some role [28]. To be considered chronic, it should last at least three months with symptoms at least three times per week; this is purposely very close to the DSM-V diagnosis. While this unified change is more restrictive than the previous DSM-IV diagnosis and limits the amount of diagnoses that can be made, it also highlights the often long-standing nature of the disorder. Unspoken in the diagnostic criteria are the many maladaptive behaviors that a person may develop in reaction to their insomnia which continue to perpetuate the issue, such as anxiety about sleep itself or nocturnal behaviors during periods of inability to sleep [27]. It is just as important to address these responses as to treat the underlying disorder as they often persist long after the precipitating incident that initially triggered the insomnia has passed.

In one of the few larger studies of adolescents with insomnia, the onset of symptoms occurred at a median age of 11 years with no difference between prepubescent boys and girls [29]. However, after puberty there is a notable uptick in the incidence of insomnia in girls, specifically a 2.75-fold increase after menarche [29], the culmination of puberty in females. Interestingly, as these young women mature into older women, while they may objectively sleep better than their male counterparts, they also report more sleep-related complaints [30]. As these women age and pass menopause, they seem to have an objective worsening of sleep architecture [30]. However, in the adolescent period alone (especially in later adolescence), there is an increase in the prevalence of insomnia in females [24]. 

## 3. Circadian Confounder

A crucial point to mention in the epidemiology of adolescent insomnia is the potential confounding diagnosis of delayed sleep wake phase disorder (DSWPD). DSWPD is characterized by a delay in the phase of the “major sleep period” with normal sleep duration and non-rapid eye movement/rapid eye movement (NREM/REM) cycling, according to the ICSD-2 classification system [31]. In the ICSD-3 criteria, these symptoms need to be ongoing for at least 3 months with objective data (e.g., sleep logs or actigraphy supported by a delayed melatonin onset) demonstrating the delay in sleep onset by more than 2 h compared to what is socially accepted or desired by the individual [27]. There also needs to be a delay in wake onset, with a significant daytime impact, again defined by the individual [27]. The ICSD-2 acknowledges the common predisposition of adolescents towards developing this syndrome [31]. No definite prevalence has been agreed upon, but between 1% and 16% of adolescents are diagnosed with DSWPD [30,32]. While behaviorally induced delayed sleep onset for reasons such as school work, employment, extracurricular activities, screen time, and parental influences [10] may certainly contribute to the development of this condition, at least one specific polymorphism has been associated with the development of DSWPD so far, and more research in this area is ongoing [33,34]. Across international studies, a consistent pattern of shifting sleep onset times has been demonstrated in adolescents, with a 1–2 h delay in weekend versus weekday bedtime [10]. This coincides with a significantly earlier school start time [10] and truncated sleep due to the lack of opportunity to sleep until the natural end of the sleep-wake phase. This leads to sleep inertia, described as an overwhelming difficulty rising upon awakening to be ready for school, work, etc.

Sleep is driven by two main processes. Process C is the circadian system, governed by the central clock, the suprachiasmatic nucleus in the hypothalamus, which dictates when in the 24 h day an individual is most likely to sleep or wake naturally [35]. Process S is the homeostatic drive for sleep, which is governed by how long an individual has been awake. This accounts for how much “sleep debt” they have accumulated [35]. These two processes partner to determine when an individual will actually fall asleep. Given the often unmet increased requirement for sleep in this age [7], there is an increased sleep debt and heightened homeostatic drive for sleep at earlier evening hours than the delayed circadian phase of the adolescent would dictate. This ability to fall asleep at these earlier times can make elucidating the diagnosis of DSWPD difficult. Since falling asleep earlier than one’s circadian time may not ensure a stable period of continuous sleep, adolescents often report intermittent difficulty with sleep onset and even maintenance, symptoms often attributed to insomnia. The treatment of insomnia (as discussed later in this article) is different than the entrainment necessary to shift one’s delayed sleep/wake phase to earlier hours [36,37,38]. Given the high prevalence of this circadian rhythm disorder, it is important to make sure it is ruled out prior to initiating a treatment for insomnia. The timing of sleep onset, wake time and quality of sleep during weekends and prolonged holidays are important to ascertain when evaluating adolescents. When permitted to sleep on a schedule that feels most “natural”, an adolescent with delayed sleep wake phase disorder will likely not experience any decrease in sleep quality, issues with sleep maintenance, or sleep inertia. An adolescent who has no issues with their sleep when following their own circadian rhythm must either find work or school schedules that accommodate their natural sleep/wake cycle or make efforts to entrain their circadian rhythm to match the requirements of their social situations. Adolescents who continue to have difficulty initiating and maintaining sleep despite sleeping according to their circadian sleep phase should undergo further evaluation for insomnia.

## 4. Impact

Insomnia is not without physiologic effects. In the adult, studies looking at “short sleepers” and “long sleepers” (≤6 h per night and >8 h per night, respectively) have shown an association with an increased risk in all-cause mortality [2]. In a large cohort study of over 3000 aging individuals with sleep time that was either too short or too long, there was a significant elevation in markers of inflammation such as interleukin-6 (IL-6) and tumor necrosis factor-alpha (TNF-α) as well as an increase in mortality in both groups compared to the mean [2]. Other studies have demonstrated decreased sleep duration being associated with obesity, metabolic syndrome, diabetes, hypertension, coronary artery disease and increased all-cause mortality in the adult population [39]. In the adolescent, fewer large cohort analyses have been performed looking at systemic consequences of decreased duration or quality of sleep. Some studies, however, have suggested an associated increase in overweight status and obesity [40,41] and an increased risk of prehypertension and hypertension independent of obesity [41,42] putting these children at risk for long term cardiovascular morbidity. Interestingly, there is no clear increased risk of sleeping too long seen in children/adolescents [41] as seen in adults.

In the adolescent suffering from sleep issues, decreased sleep quality has many deleterious effects on neurological and psychological outcomes. Along with poor performance the next day [13,14], there is also an increased risk of anxiety and depression [15], including an increased risk of suicidal ideation [16,17]. Even more frightening is that these are also young adults who are less willing to confide in a parent or seek out medical care on their own [43], allowing their sleep pathology to go unchecked. In adults, decreased sleep efficiency as well as decreased overall satisfaction with sleep have been shown to be correlated with an increased incidence of depression [39]. In teenagers, there is a strong correlation with insomnia, anxiety and depression but determining causality is challenging [44]. Although some studies have suggested that the presence of insomnia more commonly precedes the development of anxiety and depression [45,46], the consensus is that they are likely mutually influencing conditions which have many of the same underlying etiologies and that treatment of one should most certainly include addressing the others.

A large cohort of adolescents with self-described difficulty with sleep had reported daytime symptoms of chronic tiredness/fatigue and had objectively worse school performance [14]. The converse is also true, with decreased symptoms of insomnia correlating with higher grades in non-foreign language studies as well as mathematics in both boys and girls [14]. Even outside of school performance, the presence of insomnia significantly predicts school absenteeism [47]. This is likely multifactorial in that these adolescents are trying to make up for little or non-restorative sleep at night, by sleeping longer in the morning. They are either still in bed missing school or physically present in the classroom but having microsleeps when awake, similar to a drowsy driver [48]. While this latter example is extreme, it highlights a very tangible disruption in daytime alertness, when an adolescent’s main task is to stay awake in the classroom to learn.

While beyond the scope of this review, it is important to note that adolescence is a time that teenagers are getting behind the wheel, learning to drive and doing so independently for the first time. Any decrease in alertness can increase the risk of serious bodily harm, especially in the novice driver [48]. In an effort to keep adolescents safe, in school, and with the highest chance of academic success, there has been a very strong push for delaying school start times [49,50]. While this will not necessarily address every cause of sleepiness in adolescents, aligning school time with ideal periods of physiologic wakefulness in adolescents is an important step towards mitigating the risks associated with suboptimal sleep health in this population. The American Academy of Pediatrics (AAP) and American Academy of Sleep Medicine (AASM) have both put forward emphatic formal statements identifying early school start time as a key modifiable contributor to insufficient sleep in the adolescent population and suggesting that 8:30 am should be the very earliest that schools begin [49,50]. Having these recommendations from major professional organizations for the pediatric and sleep fields respectively demonstrates how vast of an issue delaying school start times has become, and will hopefully encourage more efforts at identifying and endorsing policy changes that will support cultural shifts in our society that prioritize sleep health in adolescents.

## 5. Contributing Factors & Comorbidities

Certain populations of adolescents are more vulnerable to developing insomnia than others. As previously discussed, psychopathologies such as depression and anxiety are highly correlated with insomnia. Additionally, up to 73% of children with Attention-Deficit/Hyperactivity Disorder (ADHD) endorse sleep issues pertaining to initiating and maintaining sleep [51,52]. A recent randomized controlled trial aiming to provide behavioral tools to manage sleep symptoms demonstrated improvement in the severity of the ADHD symptoms while the subjects were still taking a stimulant medication as a first line therapy [51]. There is no current data to suggest that behavioral interventions alone are as effective for ADHD as in combination with medication [45], but this remains an area to be further investigated. It has been shown that in children with ADHD, there is an increase in cortical hyperarousal, ascertained by an increase in beta frequency EEG waves resembling wake throughout the night [53]. This was seen even during the deepest stages of sleep, which should have a predominance of much slower frequencies [53]. This suggests that there is much more to be explored regarding the underlying pathophysiology of ADHD and how great of a role sleep may play in it.

Adolescents with neurodevelopmental disorders other than ADHD are also at risk for sleep difficulties [54,55]. This includes Autism Spectrum Disorders, Cerebral Palsy, Fetal Alcohol Syndrome, Down Syndrome, and other conditions manifesting early in childhood with significant cognitive and emotional/behavioral difficulties. A very recent comprehensive review on this subject reported that among children with neurodevelopmental disorders compared to typically developing controls there is an increased incidence of bedtime resistance, anxiety, difficulty settling down to sleep, delayed sleep onset, sleep maintenance issues, and overall restless sleep [54]. There are currently no specific guidelines for managing insomnia in the neurodevelopmental disability group. Likely due to this lack of guidelines, behavioral interventions for insomnia being more thoroughly studied in the typically developing cohort [56], and few pediatric patients having access to these therapies [57], an impressive majority (81%) of neurodevelopmentally impacted children has been prescribed a medication to treat insomnia in the last year [58]. Much more research is needed on the complexity of insomnia in the neurodevelopmental disability group and clarifying first line therapies and a step-wise approach to treatment.

Other populations at risk for insomnia include children with chronic illnesses [59,60,61] and in difficult social situations such as poverty and food scarcity [62] and in turbulent home environments due to an array of issues as severe as exposure to violence [62,63]. Many of these insomnia symptoms are inextricable from the PTSD, depression and anxiety that these adolescents also face [55,62].

Having another underlying sleep disorder may lead to difficulties with sleep onset and sleep maintenance. While delayed sleep wake phase disorder has a great deal of overlap with insomnia, other circadian disorders such as Irregular Sleep Wake Rhythm Disorder and Non-24 Sleep Wake Rhythm Disorder can lead to the development of similar symptoms. Although difficulties with sleep latency are not commonly seen [52], adolescents with obstructive sleep apnea (OSA) may have sleepiness during the day [64] and difficulties with sleep maintenance due to intermittent obstructive respiratory events and hypoxia [65]. Adolescents with OSA are often overweight and have difficulty maintaining attention in the classroom [65,66] symptoms that are difficult to differentiate from the adolescent with insomnia. Therefore, any concern for possible obstructive sleep apnea should be pursued with polysomnography (PSG) before attributing sleep issues to insomnia alone.

Restless legs syndrome is a disorder described by an urge to move the legs which is most prominent at rest, relieved by movement, and often presents in a circadian pattern of peak symptoms in the evening, which can greatly impact sleep onset and sleep maintenance [67]. This is a clinical diagnosis that should be inquired about when investigating complaints of insomnia, as the associated discomfort can cause a prolonged sleep onset latency. A related disorder, Periodic Limb Movement Disorder (PLMD) [27] can cause daytime sleepiness due to sleep disruption throughout the night without obvious symptoms prior to sleep onset. This diagnosis requires a polysomnogram to rule out, similar to OSA.

Finally, while narcolepsy is a disorder of central hypersomnolence, individuals suffering from this condition do report difficulties with sleep consolidation and maintenance [27]. These adolescents often present with sleep paralysis, hypnagogic/hypnapompic hallucinations and excessive daytime sleepiness, but these can be seen due to sleep deprivation from insomnia as well. Very vivid dreams and dream enactment can be seen in narcolepsy, leading to reports of difficulties staying asleep. If a suspicion for narcolepsy arises during an adolescent’s report of insomnia, it should be more thoroughly investigated and a PSG followed by a multiple sleep latency test should be considered.

## 6. Treatment

The treatment of insomnia in adults has been standardized to promote cognitive behavioral therapy for insomnia (CBT-I) as an effective and recommended initial treatment option [68]. This was reaffirmed by the American Academy of Sleep Medicine in a recent position statement on pharmacologic treatment of insomnia, as well, emphasizing that behavioral techniques need to be a standard of treatment [69]. CBT-I is comprised of a short course (4–8 weeks) of weekly visits with a trained professional to learn, implement, and follow up on techniques such as stimulus control, sleep restriction, relaxation and relapse prevention, as well as further sleep education [70]. While alone these techniques are very effective, they have shown to be more efficacious and longer lasting when put into action together with a close relationship with a CBT-I provider [70,71]. This has been found to be as efficacious and more long lasting than medication alone [71]. In the adolescent population, CBT-I delivered in person one-on-one [72] as well as in a group setting and through specific internet-based programing has been shown to be effective for the treatment of insomnia [73]. The treatment of adolescent insomnia with CBT-I was shown to decrease somatic complaints as well as anxiety, oppositional behavior, and ADHD symptoms [74]. However, similar to the adult population [70] there are currently not enough trained providers available to deliver these services to children and adolescents [75]. Until more providers become available, consideration should be given to continuing to trial group based CBT-I as well as validated internet-based CBT-I, as these are now proving to be effective for adolescents [21].

Outside of providing behavioral interventions, medications are frequently used in the pediatric population to treat insomnia, even surpassing the rates seen in the adult population [58]. In younger children, more sedating medications are used, such as off-label use of antihistamines (e.g., hydroxyzine and diphenhydramine) or supraphysiologic doses of melatonin [58]. The efficacy of melatonin may suggest that a there is a circadian misalignment contributing to/causing difficulty initiating sleep [36,37]. Interestingly, in specifically the ADHD population as well as children with neurodevelopmental disabilities, high (5 mg and above) doses of melatonin were shown to improve sleep as well as daytime behavior [76,77]. The mechanism behind this improvement is still not completely understood, other than that these children are able to get more quality hours of sleep. Fortunately, as further research continues, long term use of melatonin even at high doses has been shown to be without adverse effects [78]. Melatonin agonists (namely Ramelteon) have been approved for adolescents over 18 years old, and have been effective in small case reports with off-label use for younger children [79].

In the ADHD population, clonidine (an alpha-2 agonist) has been used as a short-acting sedating medication, to help with sleep onset in these challenging children and adolescents [80]. However, it is not without risk given its antihypertensive properties, and can cause hypotension during use and rebound hypertension after discontinuation.

While benzodiazepines were commonly used in adults for sleep onset insomnia, due to the high risk of overdose, dependency, and abuse, they are no longer commonly used [69]. An exception to this is for the treatment of parasomnias, but this is beyond the scope of this review.

The available benzodiazepine receptor agonist medications (BzRA) or “Z” drugs (Zolpidem, Zaleplon, and Eszopiclone) are medications only approved for use in patients above 18 years of age, as well. They are non-benzodiazepine medications, and augment gamma-aminobutyric acid (GABA) activity at the receptor only if GABA is present, unlike benzodiazepines which act directly at the place of the receptor even in the absence of endogenous GABA. This theoretically provides for a better safety profile [81]. Zaleplon is the shortest acting of the three, with a half-life of one hour, compared to 1.4–4.5 h for Zolpidem and 6 h for Eszopiclone [82]. In the adult population, all three have been shown to be effective in the short term for treating sleep onset and maintenance insomnia, but long-term data is lacking, with some suggestion the behavioral techniques are even more effective [71,82]. Zolpidem has been shown to have potential side effects of complex behaviors at night as well as next day “hangover” effects on driving, memory and motor ability, risks that often outweigh its benefits [83,84]. Zaleplon and Eszopiclone have been implicated in complex sleep behaviors as well, but only in limited case reports [84,85]. Zaleplon and Eszopiclone have not been studied in the pediatric population. Zolpidem has been studied in an open label trial in pediatric patients and from a pharmacokinetic perspective was shown to be well tolerated at a dose of 0.25 mg/kg [86]. However, in studies using Zolpidem off-label for sleep augmentation in specific populations such as pediatric burn victims [86] and children with ADHD [87], it was not effective for reducing sleep latency on polysomnography. There is also an increased risk of Zolpidem misuse in the adolescent population [88] given their increased propensity for impulsivity and risk-taking behavior [3]. In light of the risk profile and lack of efficacy, BzRA medications cannot be reasonably recommended for children and younger adolescents with insomnia.

Small trials of reversible orexin receptor antagonists (Suvorexant) have been shown to be relatively safe in adolescents (with abnormal dreams and daytime sleepiness as the most prominent side effects) [89]. More studies need to be performed to determine efficacy in this population.

An exciting newer medication available for adults with insomnia is a low dose of an old tricyclic antidepressant, Doxepin. In very low doses (3 mg or 6 mg) it can help with sleep maintenance and ensuring a quality duration of sleep, though not necessarily with sleep onset [90]. It has not been studied in pediatrics, but given its safety profile, consideration should be given to further studies of Doxepin in the adolescent population.

Gabapentin is an antiepileptic medication that is often utilized for the treatment of restless legs syndrome and periodic limb movement disorder when iron supplementation is not sufficient to control symptoms [67]. Aside from being sedating, it has been shown that Gabapentin also has an enhancing effect on slow wave sleep, the “deep sleep” so often sought after [91]. For this reason, it is also utilized for insomnia in that it can improve sleep maintenance [91]. In children with neurodevelopmental disorders, use of Gabapentin for insomnia (5 mg/kg to 15 mg/kg) has been shown to be effective in improving sleep based on parental report [92]. More studies are needed regarding Gabapentin use in typically developing children and adolescents, given its demonstrated efficacy and low risk profile [92].

Given the ability of antidepressant (Doxepin) and anticonvulsants (Gabapentin) medications to serve secondary roles of augmenting sleep, if a child is already on a medication for his or her psychological or physiological comorbidity, consideration should be given to examining the possible side effects of these medications, and assessing if they could be utilized to help promote better sleep.

Finally, while Trazodone is an antidepressant that has been used in the past for treating sleep difficulty in ill or neurodevelopmentally devastated children and adolescents [93,94], there is very little data on the efficacy or safety of this medication.

## 7. Conclusions & Call to Action

Adolescence is a transformative time, typified by great intellectual, physiologic, psychologic, and social changes. Insomnia deprives adolescents of the attention and cognition required to play an active role in these exciting and demanding processes. Other sleep disorders such as obstructive sleep apnea, restless legs syndrome, circadian rhythm disorders, and narcolepsy impact sleep negatively, but insomnia is more common than each of these conditions, multifactorial, and more complicated to treat [24]. Short sleep duration in the adolescent has a multitude of negative somatic, neurodevelopmental and psychological outcomes, similar to that of the adult cohorts [13,14,39]. While clear recommendations exist for treating adult insomnia with CBT-I and medication if necessary [69], treating children and adolescents is much more complex. Behavioral therapies are challenging from many angles, including gauging maturity level, respecting autonomy in patients while acknowledging parental responsibilities for decision making, as well as the paucity of sleep psychologists trained and available to provide CBT-I in younger patients [70,75]. Another layer of complexity is that a larger proportion of adolescents with physiological, psychological and developmental comorbidities report insomnia compared to the general population, and their treatment plans need to be even more individually tailored. The recent growth in the adolescent medicine and sleep medicine fields [95,96] presents an incredible opportunity to collaborate and conduct large scale analysis of adolescent sleep health, hopefully allowing us to demonstrate prevalence of insomnia among adolescents and to further analyze the predisposing, precipitating, and perpetuating factors of adolescent insomnia [70]. Perhaps the demonstration of confirmed prevalence across multiple centers, with a more rigorous exploration of the impact of poor sleep may strengthen the argument for implementation of policies such as delaying school start times. Perhaps it may simply publicize this issue and encourage more adolescents to come forward and seek treatment, or give educators/coaches/employers more credibility in suggesting to families that an adolescent may benefit from additional help. Perhaps applying systematic treatment regimens to adolescents with insomnia will allow us to standardize behavioral and pharmaceutical approaches and move towards the establishment of guidelines. One thing is clear: given the far-reaching impact of adolescent insomnia, all health practitioners who care for adolescents should remain vigilant about this important issue while making efforts to optimize their growth and development. Their future, and ours, depends on it.

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
