# Peer review of "Insomnia in Adolescence"

_medsci, 2018, doi:10.3390/medsci6030072_

Round 1

Reviewer 1 Report

This is an excellent article that addresses a knowledge gap among child and adolescent psychiatrists about pathophysiology, clinical characteristics and management strategies for Insomnia in adolescents.

The authors provided a comprehensive literature review on epidemiology, contributing factors, behavioral and pharmacological treatments for Insomnia among adolescents. This a very timely and needed manuscript that is very well written and covers all important aspects of Insomnia in youth including Future Directions and Call to Action.

I have only few minor comments to the authors.

1.       Introduction:

The following sentence “ The standard of care for insomnia has moved away from medication in recent years to more cognitive behavioral therapy for insomnia (CBT-I)29…” needs clarification that this statement pertains to adults.

2.       Epidemiology:

Please include explanations for abbreviations of: DSM-IV, DSM-V, ICSD-3.

3.       Circadian Cofounder:

Please explain abbreviation for SCN.

Most clinicians outside sleep specialty are not familiar with term” sleep inertia”. Please include definition into the text.

4.       Contributing Factors & Comorbidities:

Again, spell out “ADHD” when first used.

Author Response

thank you very much for your comments and suggestions!

This sentence has been clarified to include the caveat of age: " In adults, the standard of care for insomnia has moved away from medication in recent years to more cognitive behavioral therapy for insomnia (CBT-I) [18] ..."

2. this is an excellent point. these sources have been defined: "This suggests that it is the most common sleep disorder in this age group. These values arise from the use of the fourth edition of the Diagnostic and Statistical Manual of Mental Disorders (DSM-IV) criteria for diagnosis of primary insomnia which is defined as at least one month of symptoms including difficulties with sleep onset, maintenance, or a sensation of unrefreshing sleep [25]. This is in contrast to fifth edition of the Diagnostic and Statistical Manual of Mental Disorders (DSM-V) which has stricter criteria for insomnia diagnosis, requiring a chronicity of 3 months similar to the ICSD-3 diagnosis [26]. In the sleep field, insomnia is diagnosed according to the third edition of the International Classification of Sleep Disorders (ICSD-3);  "

3. SCN has been changed to "suprachiasmatic nucleus"

Sleep inertia is defined in the sentence that just preceds the SCN correction, "This leads to sleep inertia, described as an overwhelming difficulty rising upon awakening to be ready for school, work, etc. "

4. ADHD is now spelled out at its first appearance ". Additionally, up to 73% of children with Attention-Deficit/Hyperactivity Disorder (ADHD) endorse sleep issues pertaining to initiating and maintaining sleep [51,52].  "

thank you again for your feedback!

Reviewer 2 Report

This is a great review article targeting general pediatricians, family physicians, sleep medicine physician and any providers involved in adolescents' care. It outlines challenges and limitations of our knowledge on treated pediatric insomnia and reviews currently used treatment options. It's well written and fairly comprehensive.

Author Response

thank you so much for your kind words!